# The Combination of Predictive Factors of Pharmacokinetic Origin Associates with Enhanced Disease Control during Treatment of Pediatric Crohn’s Disease with Infliximab

**DOI:** 10.3390/pharmaceutics15102408

**Published:** 2023-09-30

**Authors:** Marla C. Dubinsky, Shervin Rabizadeh, John C. Panetta, Elizabeth A. Spencer, Annelie Everts-van der Wind, Thierry Dervieux

**Affiliations:** 1Mount Sinai Medical Center, New York, NY 10029, USA; elizabeth.spencer@mssm.edu; 2Cedars Sinai Medical Center, Los Angeles, CA 90048, USA; shervin.rabizadeh@cshs.org; 3St. Jude Children’s Research Hospital, Memphis, TN 38105, USA; carl.panetta@stjude.org; 4Prometheus Laboratories, San Diego, CA 92121, USA; aeverts@prometheuslabs.com

**Keywords:** infliximab, Crohn’s disease, pharmacokinetics, clearance, predictive factors

## Abstract

Infliximab (IFX) concentrations are a predictive factor (PF) of pharmacokinetic (PK) origin in the treatment of Crohn’s disease (CD). We evaluated Clearance, another PF of PK origin, either alone or in combination with concentrations. They were evaluated from two cohorts, the first designed to receive standard dosing (n = 37), and the second designed to proactively target therapeutic IFX concentrations (n = 108). Concentrations were measured using homogeneous mobility shift assay. Clearance was estimated using the nonlinear mixed effects methods with Bayesian priors. C-reactive protein-based clinical remission (<3 mg/L in the absence of symptoms) was used for the disease control outcome measure. Longitudinal changes in disease control due to factors including time, IFX concentration, and Clearance were analyzed using repeated event analysis. Change in objective function value (∆OFV) was calculated to compare concentration and Clearance. The results indicated that lower baseline Clearance and proactive dosing associated with enhanced disease control during induction (*p* < 0.01). Higher IFX concentrations and lower Clearance measured at the second, third, and fourth infusion yielded improved disease control during maintenance (*p* < 0.032). During maintenance, the association with disease control was better with Clearance than with concentrations (∆OFV = −19.2; *p* < 0.001), and the combination of both further minimized OFV (*p* < 0.001) with markedly improved clinical yield in the presence of both PF of PK origin. We conclude that the combination of IFX concentration and Clearance are better predictors of therapeutic outcome compared with either one alone.

## 1. Introduction

The availability of anti-tumor necrosis factor-α (TNF-α) monoclonal antibodies such as infliximab (IFX) has significantly improved the outcome of Crohn’s disease (CD), an inflammatory bowel disease (IBD) that destroys gastrointestinal tissue. For over two decades, a wealth of data have supported the notion that suboptimal pharmacokinetics (PK) associates with poor outcomes in IBD, and clinical gastroenterologists have widely adopted clinical PK and therapeutic drug monitoring (TDM) [1,2,3,4] measurements. This incorporation of individualized PK profiles in the dose decision making process informs on dose intensification strategies (DIS) [4,5] and is known to decrease the risk of treatment failure resulting from suboptimal PK, immune response against the monoclonal itself (formation of antidrug antibodies) and insufficient concentration to effectively neutralize the inflammatory burden [5,6].

For IFX, it is well-established that higher circulating IFX concentrations are associated with a higher degree of disease control across multiple immune-mediated inflammatory diseases [2,7] including rheumatoid arthritis, and the expert consensus in clinical gastroenterology practice is to implement DIS to achieve adequate exposure (e.g., >10 μg/mL immediately before the last infusion of the induction period and at least >5 μg/mL during maintenance) to sustain remission [1]. However, the association between IFX concentration and outcome is modest, and additional markers are needed to bolster the value of PK measurements.

In PK, Clearance is the volume of serum from which the drug is removed from the body as a function of time (expressed as liters per day) and an indicator of the consumption of the monoclonal antibody. Recently, IFX Clearance has emerged as a predictive factor (PF) of PK origin, both before starting therapy [8,9] and during treatment [10,11]. In the context of IFX treatment, accelerated Clearance is proposed to reflect immune response to the monoclonal antibody itself, ineffective intrinsic metabolism and recirculation of the immunoglobulin and higher inflammatory burden [12] which consumes the drug.

The high-level inflammatory burden present in CD often precludes achievement of effective concentrations on a standard dosing schedule [13]. It follows that DIS is often implemented, reactively, in the face of the symptomatic patient having suboptimal PK and having received too little drug to achieve complete neutralization of the inflammatory burden. Alternatively, DIS can be implemented proactively to ascertain dosing that achieves the minimal effective concentration to benefit both those with suboptimal PK and those with high inflammatory burden.

Several reports including observational and randomized clinical studies have established that proactive personalized dosing based on patient PK profiles associates with improved outcomes for IFX [3,14,15,16]. However, the clinical utility of proactive DIS is a matter of debate [17,18,19]. At the PK level, another key benefit of proactive dosing is that sustained IFX exposure promotes tolerance and lessens the risk of immunization against IFX antigens, as seen in patients receiving episodic IFX treatment or carriers of the HLA DQA1*05 allele [20,21,22].

In this study we evaluated the association between IFX Clearance, either alone or in combination with IFX concentrations, with disease control. Our hypothesis was that superior clinical outcomes would be achieved in the presence of higher concentrations and lower Clearance. Our results show that, in addition to IFX trough concentrations, the presence of low IFX Clearance associates with enhanced disease control, where patients with optimal PK (higher IFX trough concentrations and lower IFX Clearance) are likely to do very well on IFX.

## 2. Materials and Methods

### 2.1. Patients

The informed consent was collected from all patients enrolled in the study. The two separate cohorts of pediatric CD patients who started IFX induction were from the United States. Patients in the first cohort (Standard dosing cohort, Cedars Sinai Medical Center, SR) received standard IFX dosing (5 mg/kg given for three consecutive doses at week 0, 2 and 6 followed by maintenance treatment every 8 weeks at the same dose). Patients in the second cohort (Proactive dosing cohort, Mount Sinai Medical Center, MD, USA) received proactive dose intensification using individualized PK profiles calculated by the iDose dashboard (Baysient, LLC, Fort Myers, FL, USA) to target therapeutic concentrations (>17 µg/mL before the third induction infusion and >10 µg/mL during maintenance). All IFX samples were assayed at Prometheus Laboratories (San Diego, CA, USA) with the test results from the proactive cohort reported to the clinician within 3 days of receipt (after overnight transportation) [14].

The two cohorts were combined, with PK outcomes of the standard dosing cohort compared with those from the proactive dosing cohort.

### 2.2. Clinical PK Testing

All specimens were collected in serum separator tubes and shipped overnight to the clinical laboratory for testing (proactive dosing cohort) or stored at subzero temperature (standard dosing cohort). During induction, specimens were collected immediately before (trough) the second, third, and fourth infusion (corresponding to week 2, 6, and 14 on a standard dosing schedule or per the clinician’s recommendations in the proactive dosing cohort). During maintenance, specimens were collected at the trough before each infusion.

All IFX testing was calibrated against WHO standard (NIBSC code: 16/170) and conducted in the CLIA certified clinical PK laboratory at Prometheus Laboratories (San Diego, CA, USA). IFX and antibodies to IFX (ATI) were determined using drug-tolerant homogeneous mobility shift assay (HMSA) [23]. Albumin and C-reactive protein (CRP) levels were determined using standard immunochemistry techniques. The clinical PK parameters were estimated using a nonlinear mixed-effects modeling with Bayesian priors via Monolix (Lixoft, 2021R2, Paris, France) that incorporated amount of IFX given (in mg), weight (in kg), albumin (in g/dL), ATI (positive > 3.1 U/mL), and IFX concentration (in µg/mL) as described [24]. Clearance (expressed as L/day), the IFX-containing serum cleared from the total serum as a function of time, was calculated at each cycle; a value below 0.294 L/day (the value of the reference population) was defined as lower Clearance.

Baseline Clearance (immediately before first infusion dose) was estimated using the previously reported population PK model [24] using albumin and weight covariate estimates:Log (Baseline Clearance) = log(0.294) + 0.614 ∗ log(WT/70) − 1.2 ∗ log(ALB/4.0))

Note: WT is weight in kg and ALB is serum albumin level in g/dL.

### 2.3. Outcome Variables

The outcome variable used was the CRP-based clinical remission status achieved during the induction and maintenance periods. CRP-based clinical remission status was defined as serum CRP levels below 3 mg/L in the absence of clinical symptoms (remission), using the CRP level measured immediately before infusion. Disease activity was assessed using either the pediatric Crohn’s disease activity index (below 10 points indicates remission) in the Standard dosing cohort, or the Harvey Bradshaw index (below 5 points indicates remission) in the Proactive dosing cohort. Sustained CRP-based clinical remission status corresponded to CRP-based clinical remission status achieved at all cycles of the maintenance period.

During treatment, the longitudinal changes in PF of PK origin (concentration and Clearance) were analyzed as continuous variables, or above or below their respective thresholds. The PF of PK origin considered were IFX levels above recommended threshold achieved immediately before the second (>20 µg/mL), third (>15 µg/mL), and fourth (>10 µg/mL) infusion during induction, and above 5 µg/mL during maintenance starting at the fifth dose (week 22 on a standard dosing) either alone (the first PF of PK origin) [1] or in combination with lower Clearance (<0.294 L/day) (the second PF of PK origin).

### 2.4. Statistical Analysis

The hypothesis was that the presence of both PF of PK origin, lower Clearance and higher concentration would associate with superior disease control during treatment. Longitudinal changes in CRP-based clinical remission status (treated as repeated categorical observation) over induction and maintenance were analyzed using non-linear mixed effects modeling with a logistic regression model via Monolix (Lixoft, 2021R2, Paris, France).

Covariates including baseline Clearance, treatment group (Standard and Proactive dosing), PF of PK origin at each of the induction periods were evaluated for their impact on disease control with treatment time as regressor. The addition of a covariate to the model was considered significant (*p* = 0.05) if it decreased the −2log likelihood (−2LL) by ≥3.84 given one degree of freedom (based on the χ2 distribution).

Time to CRP-based clinical remission status achieved during treatment was estimated using standard Kaplan–Meier analysis. The hazard ratio (HR) corresponding to the ratio of the rate of CRP-based clinical remission achieved in the presence or absence of the PF of PK origin either alone or in combination. Higher HR represented improved rates of remission and thus enhanced disease control. Mann–Whitney and Fisher Exact tests were used to compare the PK parameters with the groups of patients that did and did not achieve sustained CRP-based clinical remission during maintenance. The software used for the comparison consisted of R (version 4.03), and MedCalc (MedCalc Software Ltd., Ostend, Belgium; version 20.011).

## 3. Results

A total of 145 pediatric patients were enrolled in this analysis (median age 13.3 years) and started induction (median 5 mg/kg). A total of 135 patients who started induction were also followed during maintenance with 72% of patient cycles having CRP-based clinical remission status achieved. The result summarizing the patient demographics and the PK characteristics at each of the induction cycles and during treatment is presented in Table 1.

### 3.1. Baseline Clearance and Disease Control Achieved during Induction Period

Baseline IFX clearance was 0.257 L/day (median IQR: 0.213–0.310 L/day) (Table 1). Patients from the Proactive dosing cohort had higher baseline Clearance (median 0.271 L/day; IQR: 0.227–0.327 L/day) than those from the Standard dosing cohort (median 0.234 L/day; IQR: 0.179–0.260 L/day) (*p* < 0.01) (Table 1). Longitudinal analysis with CRP-based clinical remission status achieved over the induction period (week 2, 6 and 14) revealed higher probability to achieve remission with longer time on treatment (estimate log odds: +0.042 ± 0.008 per day) (*p* < 0.001; −2LL: 470.9). Covariate analysis with the treatment group (standard dosing vs proactive dosing) and baseline Clearance revealed that higher baseline Clearance predicted reduced disease control during induction (*p* = 0.016), with proactive dosing associating with better disease control compared with Standard dosing treatment (*p* < 0.01).

The probability of achieving disease control during the induction period by treatment group and baseline Clearance (quartile analysis) is presented in Figure 1. At week 14, the probability of achieving remission was 46% with baseline Clearance value of 0.310 L/day (third quartile value) under the standard dosing schedule and increased to over 90% under the proactive dosing schedule.

### 3.2. Pharmacokinetic Parameter(s) during Induction Impact Outcomes during Maintenance

The impact of IFX concentration and Clearance measured immediately before the second, third, and fourth infusion on disease control changes achieved during maintenance is presented in Table 2. Alone, time on treatment was significantly associated with higher probability to achieve CRP-based clinical remission status.

The covariate analysis with concentrations as PF of PK origin revealed that higher IFX concentrations measured at the second (median 20.8 µg/mL IQR:14.6–30.4 µg/mL, n = 141), third (median 14.9 IQR: 7.9–23.0 µg/mL), and fourth infusion (median 10.4 µg/mL IQR: 5.7–15.0 µg/mL, n = 136) associated with higher probability of CRP-based clinical remission status achieved during maintenance (log odds = +0.045 ± 0.021, +0.114 ± 0.027 and +0.136 ± 0.032, per µg/mL, immediately before second, third, and fourth infusion, respectively) (*p* < 0.032) with significant decrease in OFV (*p* ≤ 0.020) compared with the base model (time only) (Table 2). Thus, achievement of higher exposure and concentration during induction is associated with higher probability of disease control during maintenance.

Conversely, higher Clearance measured before the second (median 0.234 L/day IQR: 0.196–0.302 L/day, n = 141), third (median 0.212 L/day IQR: 0.174–0.324 L/day, n = 141), or fourth infusions (median 0.190 L/day IQR: 0.153–0.267 L/day) all predicted worse probability to achieve CRP-based clinical remission during maintenance (log odds = −7.43 ± 2.88, −8.35 ± 2.3 and −11.90 ± 2.79 per each L/day immediately before second, third, and fourth infusion, respectively) (*p* ≤ 0.001) with longer time under treatment remaining associated with better disease control (*p* ≤ 0.012) (Table 2). We conclude that higher Clearance over the induction period negatively impacts disease control during maintenance.

The analysis with both PF of PK (IFX concentration and Clearance) as covariates (continuous covariate) in this longitudinal assessment of disease control during maintenance revealed a significant and independent association of higher concentration and lower clearance on CRP-based clinical remission where the combination of both PF of PK origin minimized the OFV as compared with the best OFV achieved with either PF of PK origin alone at the second (∆OFV = −4.7 [349.7 vs. 355.3]; *p* = 0.030) and fourth infusion (∆OFV = −6.2 [385.5 vs. 398.3]; *p* = 0.013) (Table 2). We conclude that the combination of IFX concentration and Clearance measures at each of the time points of induction are better predictors of therapeutic outcome compared with either one alone. The correlation between concentration and Clearance at each of the time points of the induction period and during maintenance is provided in Appendix A.

Time to CRP-based clinical remission status generally confirmed these findings with higher rate of remission in the presence of lower Clearance and higher concentration (Appendix A and Figure 2) during induction. Immediately before the fourth infusion, there was shorter time to remission (mean 68 ± 11 days) in the group of patients with lower Clearance (<0.294 L/day) compared with those with higher Clearance (188 ± 35 days) (HR = 2.1 95%CI: 1.3–3.2) (*p* < 0.001) and similar results were observed with higher concentrations (>10 µg/mL vs. <10 µg/mL 53 ± 12 vs. 142 ± 21 days, respectively) (HR = 2.1 95%CI 1.3 to 3.2) (*p* < 0001).

The impact of the presence of both PF of PK origin during induction (Clearance below 0.294 L/day and IFX concentrations above 20 µg/mL, 15 µg/mL, and 10 µg/mL immediately before the second, third, and fourth infusion, respectively) on time to remission is presented in Figure 3. Immediately before the fourth infusion there was shorter time to CRP-based clinical remission status achieved in the presence versus the absence of both PF of PK origin (mean 34 ± 10 days and 181 ± 41 days, respectively) (HR = 3.1 95%CI; 1.8–5.4) (*p* < 0.001). Results were more nuanced with the measures collected before the second infusion (*p* = 0.126) and significant at the third infusion (*p* < 0.001).

There was also a significant impact of PK measurements on sustained CRP-based clinical remission during maintenance (remission achieved at all maintenance cycles for any given patient) (Table 3). Immediately before the fourth infusion, there was lower Clearance in the group of patients who sustained remission during their maintenance compared with the group who did not (median 0.175 L/day IQR: 0.132–0.214 L/day vs. 0.247 L/day IQR: 0.167–0.313 L/day) (*p* < 0.001). Similarly, there were higher concentrations immediately before the fourth infusion in the group who did vs. the group who did not sustain remission during maintenance (median 13.0 µg/mL IQR: 8.7–18.8 µg/mL vs. 7.8 µg/mL IQR: 2.2–11.9 µg/mL, respectively (*p* < 0.001). The presence of both PF of PK origin measured immediately before the second, third, and fourth infusion associated with higher proportion of patients achieving sustained disease remission during maintenance (49% [32/65], 60% [40/67], and 64% [43/67], respectively) compared with the absence of both PF of PK origin (27% [16/55], 27% [17/63], and 26% [17/64], respectively) (*p* < 0.01). Odds ratio analysis revealed that there was a 2.6- (95%CI: 1.2–5.6) (*p* = 0.017), 4.0- (95%CI: 1.9–8.5) (*p* < 0.001) and 5.0-fold (95%CI: 2.3–10.5) (*p* < 0.001) higher likelihood of sustained CRP-based clinical remission in the presence, versus the absence, of both PF of PK origin at the second, third and fourth infusion of the induction period, respectively. Thus, the presence of both PF of PK origin during induction predicts improved disease outcome and control during maintenance.

### 3.3. Pharmacokinetic Parameter(s) during Maintenance Impacts Outcomes

The impact of IFX concentration, Clearance, and time on treatment during maintenance (starting week 22) on the longitudinal change in CRP-based clinical remission status achieved is presented in Table 4. Higher IFX concentration and lower Clearance associated with improved outcomes (log odds = +0.12 ± 0.027 per µg/mL and −16.71 ± 2.28 per L/day, respectively (*p* < 0.001) with Clearance having significantly stronger effect than concentration with lower OFV (380.8 vs. 400.0; ∆OFV = −19.2; *p* < 0.001). In addition, the combination of IFX concentration and Clearance outperformed either one alone (∆OFV = −19.0 [400 vs. 371.0; *p* < 0.001] versus concentration alone and ∆OFV = −10.8 [380.8 vs. 371.0; *p* < 0.001] versus Clearance alone). We conclude that the combination of IFX concentration and Clearance are better predictors of therapeutic outcome compared with either one alone.

A total of 419 cycles with CRP-based clinical remission status were collected in the group of 135 patients who entered maintenance. Overall, there was a greater proportion of maintenance cycles with both PF of PK origin (lower Clearance [<0.294 L/day] and higher concentration [>5 µg/mL]) in the group of patients enrolled in the Proactive dosing cohort (79%, 236/299) compared with the Standard dosing cohort (42%, 50/120) (*p* < 0.01). There was also a greater proportion of patients who achieved sustained CRP-based clinical remission in the Proactive dosing cohort (58% [60/103]) compared with the Standard dosing cohort (31% [10/32]; *p* = 0.007).

Repeated event analysis during maintenance revealed that the presence of both PF of PK origin (concentration >5 µg/mL with Clearance <0.294 L/day) yielded enhanced disease control (*p* < 0.001) with lower OFV compared with IFX concentration above 5 µg/mL alone (∆OFV = −12.6 [361.6 vs. 349.0]; *p* < 0.001) or Clearance below 0.294 L/day alone (∆OFV = −4.7 [353.7 vs. 349.0]; *p* = 0.030). Finally, the presence of both PF of PK origin immediately before the first maintenance infusion (concentration > 10 µg/mL with Clearance below 0.294 L/day) further produced additional clinical benefit and minimization of OFV (∆OFV = −10.6; [349.0 vs. 338.4]; *p* < 0.001) (Appendix A and Figure 4). After 32 weeks of maintenance therapy, the probability to achieve CRP-based clinical remission was only 44% in the absence of both PF of PK origin at the end of the induction period and during maintenance and reached 95% in the presence of both PF of PK origin.

## 4. Discussion

This report evaluated predictive factors (PF) of pharmacokinetic (PK) origin, lower Clearance and higher trough concentrations, either alone or in combination, as indicators of achieving optimal PK and disease control in pediatric patients with CD starting treatment with IFX. Our rationale was that lower Clearance, and thus effective retention of IFX to neutralize inflammatory burden would associate with improved disease control, the opposite of what is observed when accelerated Clearance results from active disease that consumes IFX and worsens in the presence of immunization against IFX [25,26,27].

This was a retrospective exploratory analysis and our data suggested enhanced disease control achieved during induction in the group of patients undergoing Proactive dosing, thereby adding to the body of evidence that this DIS (by model-informed precision dosing tools [iDose] coupled with HMSA assay) has strong clinical utility compared with Standard dosing [11,14]. This conclusion was based on the higher probability of achieving remission during the induction period (log odds: 2.53 ± 0.61, *p* < 0.001) and the greater proportion of patients with sustained disease remission (53% vs 31%) in the group treated with proactive dosing.

We noticed that higher baseline Clearance associated with worse induction outcomes as already reported [9] (log odds = −5.77 ± 2.40 per L/day). Our results also indicated that the negative impact of higher Clearance on outcome was minimized by the implementation of DIS and proactive dosing (log odds = +2.53 ± 0.61). We acknowledge that the absence of a comparator group undergoing reactive DIS and the lack of a randomized control group does not allow us to draw conclusions on the superiority of proactive dosing versus other DIS as seen with other IFX [3,15] and adalimumab [28] reports. Yet, the data suggest that enhanced disease control was achieved in the proactive dosing group that followed optimal management of dosing and exposure commensurate with inflammatory burden [29,30,31,32].

As expected, higher IFX concentrations, measured during induction, predicted improved outcomes during maintenance. Immediately before the second, third, and fourth infusion patients presenting with IFX concentrations above 20, 15, and 10 µg/mL, respectively, (expert recommended cutoffs) had higher rates of remission than those with concentrations below cutoffs (HR = 1.4 95%CI: 0.9 to 2.1; HR = 2.3 95%CI: 1.5 to 3.6; HR = 2.2 95%CI: 1.4 to 3.3, respectively). These data add to the body of evidence that optimal exposure achieved early is important for patient outcomes [1]. Moreover, lower Clearance (<0.294 L/day) during induction also predicted better disease control (HR = 1.3 95%CI: 0.9 to 2.0, HR = 1.5 95%CI: 1.0 to 2.3; and HR = 2.1 95%CI: 1.3 to 3.2 at the second third and fourth infusion, respectively), and these data are consistent with the notion that sufficient neutralization of TNF-α allows greater retention of IFX in the central compartment, the opposite of what is observed in the presence of higher inflammatory burden and tissue inflammation that serves as a sink for IFX with attendant accelerated Clearance, as reported [6].

In combination, lower Clearance and higher drug concentrations during induction yielded improved disease control during maintenance (log Odds = +0.087 ± 0.029 and −9.14 ± 2.67, at the fourth infusion, respectively), where the objective function value (OFV) was minimized compared with concentration and Clearance alone, thus suggesting that the measurement of Clearance might provide additional value in clinical practice, where further clinical improvements can be expected in the presence of higher concentration and lower Clearance.

This was also confirmed in maintenance, where Clearance was superior to concentration (lower OFV, ∆OFV = −19.2) and the combination of Clearance and concentration together outperformed either one alone (∆OFV = −19.0 versus concentration alone and ∆OFV = −10.8 versus Clearance alone). Time to CRP-based remission and Kaplan–Meier analysis confirmed these findings with higher rates of remission in the presence versus the absence of both PF of PK origin at the second, third and fourth infusion (HR = 1.8 95%CI: 1.0 to 3.2, HR = 2.5 95%CI: 1.5 to 4.1, and HR = 3.1 95%CI: 1.8 to 5.4, respectively).

These data support the assertion that Clearance is as good, if not better, than concentration in associating with clinical outcomes, likely due to an underlying degree of causality with improvements in clinical benefit; this supports the clinical utility of PK measures for patients with IBD [3,14,33]. While it is important to validate this concept in additional cohorts of patients receiving IFX, this finding of better outcomes in a group of patients with improved IFX retention, as indicated by lower Clearance and higher exposure, is novel. It is, further, tempting to suggest that the combination of these PF of PK origin may also associate with enhanced disease control for other monoclonal antibodies as we anticipate that the lower elimination of other monoclonal antibodies from the central compartment produces optimal PK and thus higher likelihood of disease control, as seen with IFX in this study. We acknowledge that endoscopy was not available in this cohort of pediatric patients with CD, and it is important to also evaluate the impact of PF of PK origin on endoscopic remission.

Our data show that in the presence of both PF of PK origin, at the end of the induction period and during maintenance, superior disease control is achieved. After 32 weeks of maintenance therapy, the probability of achieving CRP-based clinical remission was 44% in the absence of both PF of PK origin and reached 95% in the presence of both PF of PK origin. It follows that optimizing IFX exposure early during induction, to reach and then sustain exposure commensurate with the effective neutralization of inflammatory burden, is vital to achieving and maintaining remission. While prospective studies are important, our findings support that the management of the immune-mediated inflammatory disease patient can be improved through both the avoidance of negative outcomes associated with a failure to maintain sufficient drug concentrations and the maximization of their sustained remission through the optimization of clearance and concentration.

As we enter this new era of precision-guided dosing, DIS can be implemented reactively or proactively with the same goal of remediating underexposure and insufficient control of inflammation. We are not ignoring the fact that these hypotheses-generating data are subject to type 1 error and the rejection of the null hypothesis that benefit PK and Clearance could be happening by chance. While the group of patients presenting with the PF of PK origin, and thus with optimal PK, appear to be doing very well under IFX, it is also evident that some of these will not respond to IFX, owing to an unsuitable mechanism of action.

Additional cohorts are important to further evaluate the impact of PF of PK origin on outcome; however, Clearance has been shown to be particularly important in pediatrics, making these data important contributions to knowledge in pediatric IBD. Importantly, the currently active OPTIMIZE trial [30] (NCT04835506) is exploring the benefit of proactive DIS compared with standard of care in a randomized controlled trial of patients 16 and older, which will further enhance the knowledge that we have presented herein.

## Figures and Tables

**Figure 1 pharmaceutics-15-02408-f001:**
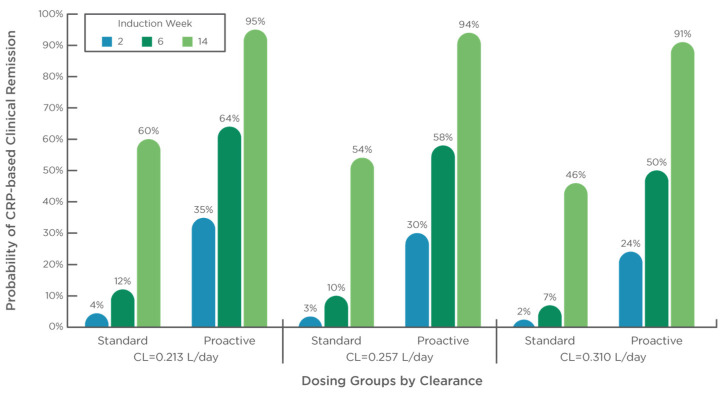
Baseline Clearance and dosing strategy impact CRP-based clinical remission during induction. Probability to achieve CRP-based clinical remission status by week (2, 6, and 14) during induction by treatment group (standard and proactive) and baseline IFX Clearance (median, with interquartile range). Estimates are: θ_pop_ = 2.49 ± 0.78 (*p* = 0.001); θ_cl_ = −5.77 ± 2.40 per L/day (*p* = 0.016); θ_proactive_ = +2.53 ± 0.61 (*p* < 0.001); θ_time_ = + 0.042 ± 0.0064 per day (*p* < 0.001).

**Figure 2 pharmaceutics-15-02408-f002:**
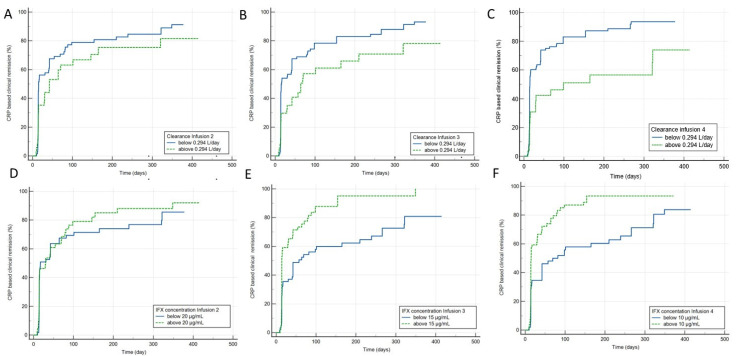
Higher IFX concentrations and lower Clearance during induction associate with CRP-based clinical remission status. (**A**) Clearance below or above 0.294 L/day, infusion 2 (HR = 1.3 95%CI: 0.9 to 2.0; *p* = 0.188); (**B**) Clearance below or above 0.294 L/day, infusion 3 (HR = 1.5 95%CI: 1.0 to 2.3; *p* = 0.047); (**C**) Clearance below or above 0.294 L/day, infusion 4 (HR = 2.1 95%CI: 1.3 to 3.2; *p* = 0.001); (**D**) IFX concentrations above or below 20 μg/mL, infusion 2 (HR = 1.4 95%CI: 0.9 to 2.1; *p* = 0.140); (**E**) IFX concentrations above or below 15 μg/mL, infusion 3 (HR = 2.3 95%CI: 1.5 to 3.6; *p* < 0.001); (**F**) IFX concentrations above or below 10 μ/mL, infusion 4 (HR = 2.2 95%CI: 1.4 to 3.3; *p* < 0.001); estimates are provided in Appendix A.

**Figure 3 pharmaceutics-15-02408-f003:**
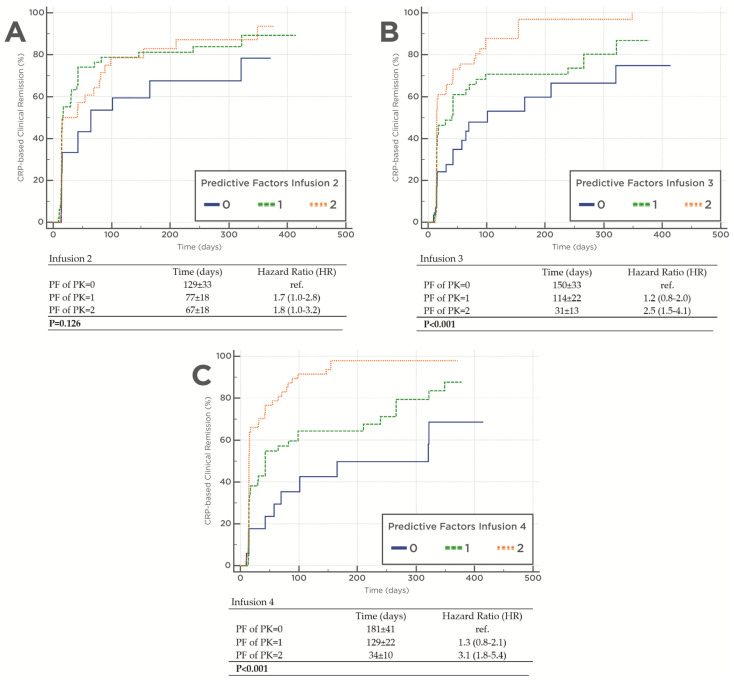
Predictive factors of pharmacokinetic origin in association with CRP-based clinical remission status. (**A**) PF of PK origin infusion 2; (**B**) PF of PK origin infusion 3; (**C**) PF of PK origin infusion 4.

**Figure 4 pharmaceutics-15-02408-f004:**
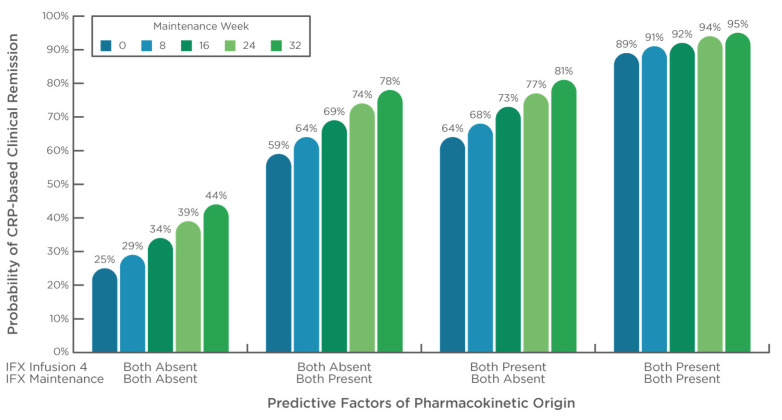
PF of PK origin during maintenance therapy and CRP-based clinical remission status. Repeated event analysis and probability of CRP-based clinical remission status over the maintenance period (>4 infusions) stratified by the presence or absence of both PF of PK origin determined immediately before the fourth infusion (IFX concentrations above 10 µg/mL with Clearance below 0.294 L/day) and during maintenance (IFX concentrations above 5 µg/mL with Clearance below 0.294 L/day). The probability of remission is given over 32 weeks of maintenance treatment. Week 0 corresponds to the start of maintenance. Estimates: θ_pop_ = 0.35 ± 0.50 (*p* = 0.012); θ_time_ = 0.002 ± 0.001 (*p* = 0.046) per day (*p* < 0.001); θ_both PF of PK_ 4th _inf_ = 2.44 ± 0.64 (*p* < 0.001); θ_both PF of PK_ = 2.51 ± 0.73 (*p* < 0.001).

**Table 1 pharmaceutics-15-02408-t001:** Patient characteristics during induction and during maintenance therapy. Results are reported as median, interquartile range (IQR) or % as appropriate.

	Standard Dosing	Proactive Dosing	Overall
Pre-infusion 1 (patient number)	37	108	145
Gender (female)	45.9% (17/37)	44.4% (48/108)	44.8% (65/145)
Age (years)	13.0 [10.0–14.0]	13.5 [11.2–15.2]	13.3 [11.0–15.1]
Immunosuppressants	21.6% (8/37)	7.4% (8/108)	11.0% (16/145)
Dose (mg/kg)	5.0 [5.0–5.0]	5.0 [5.0–10.0]	5.0 [5.0–5.2]
Weight (kg)	39.1 [29.5–52.9]	41.5 [30.3–53.0]	41.5 [29.9–52.9]
Clearance baseline (L/day)	0.234 [0.179–0.260]	0.271 [0.227–0.327]	0.257 [0.213–0.310]
Albumin (g/dL)	4.0 [3.5–4.3]	3.2 [2.8–3.7]	3.4 [2.9–3.9]
CRP-based clinical remission	8.1% (3/37)	14.8% (16/108)	13.1% (19/145)
Pre-infusion 2 (patient number)	36	105	141
Time (days)	14.0 [14.0–14.0]	14.0 [13.1–14.2]	14.0 [13.3–14.0]
Dose (mg/kg)	5.0 [5.0–5.0]	5.1 [5.0–9.9]	5.0 [5.0–6.0]
Weight (kg)	40.1 [30.7–52.9]	41.3 [31.3–54.0]	40.8 [31.0–54.0]
Albumin (g/dL)	3.9 [3.7–4.2]	3.6 [3.3–3.9]	3.7 [3.4–4.0]
Clearance (L/day)	0.208 [0.194–0.249]	0.243 [0.200–0.308]	0.234 [0.196–0.302]
IFX (µg/mL)	16.2 [10.4–27.5]	22.0 [15.4–32.1]	20.8 [14.6–30.4]
ATI positive status	5.4% (2/37)	0.0% (0/108)	1.4% (2/145)
CRP-based clinical remission	33.3% (12/36)	70.5% (74/105)	61.0% (86/141)
Pre-Infusion 3 (patient number)	35	106	141
Time (days)	42.0 [42.0–42.0]	38.0 [32.3–42.0]	42.0 [34.9–42.0]
Dose (g/kg)	5.0 [5.0–5.0]	5.1 [5.0–10.0]	5.0 [5.0–9.9]
Weight (kg)	41.1 [30.9–54.4]	42.6 [32.8–54.8]	42.4 [32.7–54.9]
Albumin (g/dL)	4.1 [3.8–4.4]	3.9 [3.7–4.1]	3.9 [3.7–4.2]
Clearance (L/day)	0.213 [0.170–0.405]	0.212 [0.174–0.300]	0.212 [0.174–0.324]
IFX (µg/mL)	7.4 [1.3–21.3]	15.5 [10.8–24.2]	14.9 [7.9–23.0]
ATI positive status	8.3% (3/36)	6.7% (7/105)	7.1% (10/141)
CRP-based clinical remission	40.0% (14/35)	69.8% (74/106)	62.4% (88/141)
Pre-Infusion 4 (patient number)	32	104	136
Time (days)	98.0 [98.0–98.0]	74.2 [63.0–90.9]	83.4 [66.1–98.0]
Dose (mg/kg)	5.0 [5.0–5.0]	10.0 [6.9–10.0]	9.9 [5.0–10.0]
Weight (kg)	42.2 [32.0–54.8]	44.9 [34.0–56.2]	43.9 [33.5–55.9]
Albumin (g/dL)	4.0 [3.7–4.2]	4.0 [3.8–4.2]	4.0 [3.7–4.2]
Clearance (L/day)	0.215 [0.149–0.262]	0.188 [0.155–0.269]	0.190 [0.153–0.267]
IFX (µg/mL)	4.5 [1.3–9.2]	12.2 [7.8–17.7]	10.4 [5.7–15.0]
ATI positive status	28.6% (10/35)	0.9% (1/106)	7.8% (11/141)
CRP-based clinical remission	43.8% (14/32)	74.0% (77/104)	66.9% (91/136)
Maintenance (patient number)	32	103	135
Number of cycles	120	299	419
Dose (mg/kg)	5.0 [5.0–5.0]	10.0 [9.9–10.0]	9.9 [5.0–10.0]
Weight (kg)	43.4 (34.1–56.5)	46.0 (34.8–59.5)	45.5 (34.7–58.6)
Albumin (g/dL)	3.9 (3.7–4.2)	4.0 (3.9–4.2)	4.0 (3.8–4.2)
Clearance (L/day)	0.185 (0.134–0.244)	0.200 (0.160–0.262)	0.192 (0.154–0.253)
IFX (µg/mL)	4.4 (1.1–6.8)	12.4 (8.3–17.9)	10.0 (5.2–15.9)
ATI positive status	26.7% (32/120)	4.7% (14/299)	11.0% (46/419)
CRP-based clinical remission	60% (72/120)	76.9% (230/299)	72.0% (302/419)
Sustained CRP-based clinical remission	31% (10/32)	58% (60/103)	52% (70/135)

**Table 2 pharmaceutics-15-02408-t002:** Induction PK parameters immediately before the second, third and fourth infusions impact CRP-based clinical remission status during maintenance.

	Parameter *	Second Infusion Estimates	Third Infusion Estimates	Fourth Infusion Estimates
Time only (days)	θ_pop_	0.75 ± 0.45 (*p* = 0.096)	0.68 ± 0.43 (*p* = 0.114)	0.35 ± 0.47 (*p* = 0.456)
	θ_time_	0.004 ± 0.001 (*p* < 0.001)	0.004 ± 0.002 (*p* = 0.003)	0.006 ± 0.002 (*p* = 0.003)
	−2LL	360.7	413.7	412.5
Time (days) and IFX Concentrations (µg/mL)	θ_pop_	0.185 ± 0.667 (*p* = 0.782)	−1.33 ± 0.56 (*p* = 0.0018)	−1.02 ± 0.56 (*p* = 0.069)
	θ_concentration_	0.045 ± 0.021 (*p* = 0.032)	0.114 ± 0.027 (*p* < 0.001)	0.136 ± 0.032 (*p* < 0.001)
	θ_time_	0.004 ± 0.001 (*p* < 0.001)	0.004 ± 0.001 (*p* < 0.001)	0.005 ± 0.001 (*p* < 0.001)
	−2LL	355.3 (∆ = −5.4; *p* = 0.020)	389.7 (∆ = −24.0; *p* < 0.001)	398.3 (∆ = −14.2; *p* < 0.001)
Time (days) and Clearance (L/day)	θ_pop_	+2.5 ± 0.96 (*p* = 0.009)	+2.77 ± 0.80 (*p* = 0.001)	+3.11 ± 0.83 (*p* < 0.001)
	θ_CL_	−7.43 ± 2.88 (*p* = 0.001)	−8.35 ± 2.31 (*p* < 0.001)	−11.90 ± 2.79 (*p* < 0.001)
	θ_time_	+0.005 ± 0.002 (*p* = 0.012)	+0.004 ± 0.002 (*p* = 0.012)	+0.005 ± 0.002 (*p* < 0.001)
	−2LL	354.4 (∆ = −6.3; *p* = 0.012)	399.7 (∆ = −14.0; *p* < 0.001)	391.7 (∆ = −20.8; *p* < 0.001)
Time (days), IFX concentration (µg/mL) and Clearance (L/day)	θ_pop_	−1.48 ± 1.04 (*p* = 0.155)	−0.07 ± 1.08 (*p* = 0.94)	+1.49 ± 0.93 (*p* = 0.109)
	θ_concentration_	+0.038 ± 0.021 (*p* = 0.074)	+0.092 ± 0.030 (*p* = 0.002)	+0.087 ± 0.029 (*p* = 0.003)
	θ_CL_	−6.33 ± 2.74 (*p* = 0.003)	−3.37 ± 2.34 (*p* = 0.015)	−9.14 ± 2.67 (*p* = 0.015)
	θ_time_	0.046 ± 0.009 (*p* = 0.021)	+0.004 ± 0.002 (*p* = 0.012)	+0.004 ± 0.002 (*p* = 0.001)
	−2LL	349.7 (∆ = −11.0; *p* = 0.001)	387.4 (∆ = −26.3; *p* < 0.001)	385.5 (∆ = −27.0; *p* < 0.001)

* Model: logit (Probability of CRP-based Remission) = θ_pop_ + θ_covi_ ∗ cov_i_ + ….

**Table 3 pharmaceutics-15-02408-t003:** IFX Clearance and concentrations during induction impact sustained CRP-based clinical remission during maintenance. Sustained CRP-based clinical remission corresponds to CRP below 3 mg/L in the absence of symptoms at all maintenance cycles post-induction.

	Predictive Factor, Clearance		Predictive Factor, IFX Concentrations	
Pre-Infusion	L/day	Below Cutoff ^a^	µg/mL	Above Cutoff ^b^
** Infusion 2 **				
Not sustained	0.259 [0.205–0.317]	65% (35/55)	17.0 [12.6–23.1]	33% (18/55)
Sustained	0.221 [0.194–0.266]	77% (50/65)	26.1 [17.0–34.9]	61% (40/65)
*p* Value	*p* = 0.025	*p* = 0.110	*p* < 0.001	*p* = 0.001
** Infusion 3 **				
Not sustained	0.241 [0.188–0.399]	66.1% (41/63)	12.1 [5.0–18.6]	36% (23/63)
Sustained	0.187 [0.154–0.239]	85.0% (56/67)	20.7 [12.7–31.1]	67% (45/67)
*p* Value	*p* < 0.001	*p* = 0.154	*p* < 0.001	*p* < 0.001
** Infusion 4 **				
Not sustained	0.247 [0.167–0.313]	62.5% (40/64)	7.8 [2.2–11.9]	39% (25/64)
Sustained	0.175 [0.132–0.214]	97.1% (65/67)	13.0 [8.7–18.8]	67% (45/67)
*p* Value	*p* < 0.001	*p* < 0.001	*p* < 0.001	*p* < 0.001

^a^ <0.294 L/day; ^b^ Infusion 2: >20 μg/mL; Infusion 3: >15 μg/mL; Infusion 4: >10 μg/mL.

**Table 4 pharmaceutics-15-02408-t004:** Clearance and concentrations during maintenance impact CRP-based clinical remission during maintenance.

	Parameter *	Estimates
Time only (days)	θ_pop_	+0.65 ± 0.46 (*p* = 0.158)
	θ_time_	+0.0047 ± 0.0014 (*p* < 0.001)
	−2LL	424.8
Time (days) and IFX Concentrations (µg/mL)	θ_pop_	−0.84 ± 0.63 (*p* = 0.312)
	θ_time_	+0.0055 ± 0.0018 (*p* < 0.001)
	θ_concentration_	+0.120 ± 0.027 (*p* < 0.001)
	−2LL	400.0 (∆ = −24.8; *p* < 0.001)
Time (days) and Clearance (L/day)	θ_pop_	+4.05 ± 0.57 (*p* < 0.001)
	θ_time_	+0.0055 ± 0.0017 (*p* < 0.001)
	θ_CL_	−16.71 ± 2.28 (*p* < 0.001)
	−2LL	380.8 (∆ = −44.0; *p* < 0.001)
Time (days), IFX concentration (µg/mL) and Clearance (L/day)	θ_pop_	+1.98 ± 0.56 (*p* < 0.001)
	θ_time_	+0.0058 ± 0.0016 (*p* < 0.001)
	θ_concentration_	+0.093 ± 0.025 (*p* < 0.001)
	θ_CL_	−12.84 ± 2.27 (*p* < 0.001)
	−2LL	371.0 (∆ = −53.8; *p* < 0.001)

* Model: logit (Probability of CRP-based remission) = θ_pop_ + θ_covi_ ∗ cov_i_+….

## Data Availability

No data sharing available.

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
