# Peer review of "The Combination of Predictive Factors of Pharmacokinetic Origin Associates with Enhanced Disease Control during Treatment of Pediatric Crohn’s Disease with Infliximab"

_pharmaceutics, 2023, doi:10.3390/pharmaceutics15102408_

Round 1

Reviewer 1 Report

The study titled "Predictive Factors of Pharmacokinetic Origin Combined for Enhanced Disease Control in Pediatric Crohn’s Disease Treated with Infliximab" investigates the role of infliximab (IFX) concentrations and clearance, both predictive factors (PF) of pharmacokinetic (PK) origin, in the treatment of Crohn's disease (CD) in pediatric patients. The study includes two cohorts: one receiving standard dosing (n=37) and the other proactively targeting therapeutic IFX concentrations (n=108). IFX concentrations were measured using a homogenous mobility shift assay, and clearance was estimated using advanced statistical methods. Disease control, based on C-reactive protein levels (<3mg/L in the absence of symptoms), was the primary outcome.

Key findings include, 1. Lower baseline clearance and proactive dosing were associated with improved disease control during the induction phase (p<0.01); 2. Higher IFX concentrations and lower clearance at the second, third, and fourth infusions were linked to better disease control during the maintenance phase (p<0.032); 3. During maintenance, clearance demonstrated a stronger association with disease control compared to concentrations (∆OFV= -19.2; p<0.001); 4. Combining both IFX concentration and clearance significantly improved clinical outcomes (p<0.001). In conclusion, the study suggests that focusing on PF of PK origin and achieving lower clearance during both induction and maintenance phases enhances disease control in pediatric CD patients treated with IFX. Furthermore, using a combination of IFX concentration and clearance proves to be a more reliable predictor of therapeutic success compared to either factor alone.

I appreciate the authors effort on this study, however, following comments should be addressed before publication,

1.       The background section is generally well-structured and clear. It progresses logically from the introduction of the treatment (IFX) to the importance of individualized dosing strategies. However, consider adding a brief introductory sentence to set the stage and provide context for readers unfamiliar with CD and IFX.

2.       When introducing concepts like "Clearance," it would be beneficial to provide a brief explanation for readers who may not be well-versed in PK terminology. Explain what Clearance means in the context of IFX treatment.

3.       In the middle of the section, there is a mention of "too little and too late" issues. While this phrase conveys a sense of urgency, it could benefit from a more explicit explanation of what these issues entail. This will help readers understand the problem statement better.

4.       the authors conclude the background section by mentioning that your study evaluated the association between Clearance and disease control. However, you may want to briefly state the key findings or hypotheses you intend to test in your study. This can give readers a preview of what to expect.

5.       The methodology section is generally clear, but there are a few instances where clarity could be improved. For example, in paragraph 2.1, you mention "Proactive dosing cohort, PRECISION IFX trial [NCT02624037]," which might benefit from a brief explanation of what the PRECISION IFX trial is or why it's relevant to your study.

6.       In paragraph 2.4, the authors describe the statistical analysis methods used, which is good. However, you may want to provide a bit more detail on the logistic regression model, especially if you used specific covariates. Also, consider specifying the statistical software used for analysis.

7.       When describing the Kaplan-Meier analysis, the authors mention "specific PF of PK origin." Consider explaining what these specific PFs are or referring to where they were previously defined.

8.       While you present associations between IFX concentration, Clearance, and disease control, you could also briefly discuss the clinical implications of these findings. What do they suggest for treatment strategies or patient management?

9.       Consider briefly mentioning any limitations of the study in the Results section, even though a more detailed discussion is typically reserved for the Discussion section.

10.   The current version requires proofreading to double-check the typos and grammar or punctuation issues, to enhance overall readability.

The current version requires proofreading to double-check the typos and grammar or punctuation issues, to enhance overall readability.

Author Response

  1. We have followed the recommendation of the reviewer and provided an introduction sentence for context.
  2. We have followed the recommendation of the reviewer and provided a definition of Clearance in the revised manuscript.
  3. We have followed the recommendation of the reviewer and provided edited the sentence in questions to provide clarity.
  4. We have followed the recommendation of the reviewer and stated the hypothesis and key findings.
  5. We have followed the recommendation of the reviewer and provided edits.
  6. We have followed the recommendation of the reviewer and provided edits and detailed the methods with equations used in the analysis.
  7. We have deleted the word specific and describe the PF of PK origin used in the study.
  8. We have added statement in the discussion regarding treatment strategies and patient management.
  9. We respectfully believe that the result section should not discuss the limitations of the study.
  10. We have proofread the document and verified the grammar.

Reviewer 2 Report

The manuscript titled “The Combination of Predictive Factors of Pharmacokinetic Origin Associates with Enhanced Disease Control During Treatment of Pediatric Crohn’s Disease with Infliximab” aims to demonstrate that Infliximab (IFX) concentrations are indeed a predictive factor of pharmacokinetic origin in the treatment of Crohn’s disease.

The manuscript provides evidence to support such aims, but the manuscript has to be cleaned up and address major issues with the results.

For instance, the heads in the abstract are not needed. The tables and figures should follow when first mentioned in the text and just put one after the other at the end.

There are minor issues with spacing, consistency, or other English languages issues that needs to be fixed.

An example is

 *(TDM)[1-3] or (DIS) [4,5]. There should be no space before the citations.

*placement of punctuation is wrong “diseases.[2, 7]”

*verb tense should be past “patient PK profiles associates with improved outcomes

*Also verb tense is wrong here, should be past tense. Our results show that

Difficult language that needs improvement

*All patients were consented

Following use of brackets is wrong.(Proactive dosing cohort, PRECISION IFX trial [NCT02624037]) received” Also what is PRECISION IFX trial [NCT02624037]?  Is it the name of the research or what? And what is NCT….. is it the ethical statement decision number?  There are other spots in the manuscript that use the wrong brackets as well. The whole manuscript needs to be checked and corrected.

Punctuation is needed at the end of the sentence

PK model[24] using albumin and weight

Citations are needed for these statements:

*CRP-based clinical remission 102 status was defined as CRP levels below 3 mg/L in the absence of disease activity (remis-103 sion), as determined using the CRP level immediately before infusion.

*The addition of a co-119 variate to the model was considered significant (p=0.05) if it decreased the -2log likelihood 120 [-2LL] by ≥ _3.84 given one degree of freedom (based on the 2 distribution).

*PF of PK origin where higher HR represented improved rates of remission and thus en-126 hanced disease control.

Results

As per author’s guidelines: Results: Provide a concise and precise description of the experimental results, their interpretation as well as the experimental conclusions that can be drawn.

The reviewer finds that a concise and precise description of the results is not provided in the results section. For example, only one sentence is provided for Table 1 “Patient demographics across the two cohorts are reported in Table I.” and again for the following results “Time to CRP-based clinical remission status generally confirmed these findings with 161 higher rate of remission with lower Clearance and higher concentration (supplementary 162 Table S1 and Figure S2) during induction.

Also when results are provided, the reader should know exactly where to find the result from the text. For example, the following does not direct the reader where the result is “As expected, repeated event analysis 136 over the induction period revealed that disease control increased with time under treat-137 ment (log odds: +0.042±0.008 per day) (p<0.001; -2LL: 470.9).”  and another example as “Accounting for time, treatment group and baseline Clearance, higher 141 baseline Clearance predicted reduced disease control during induction (log odds: -142 5.77±2.4 per each L per day) (p=0.016), and Proactive dosing treatment resulted in better 143 disease control compared to Standard dosing (log odds: 2.53±0.61, p<0.001”.  The reviewer will not outline all the statements as it is a must that the authors go line by line and improve the result section.

In addition, it is a must that the tables and figures have legends so that they stand on their own. Meaning that one understands the tables and figures in their entirely because the legend provides enough description.

*There is no mention of figure 2 in the text and there is not Figure S2. Why?

*Spacing and punctuation issues:  Estimates are: pop=2.49±0.78 (p=0.001); cl=-5.77±2.40 283 per L/day (p=0.016); proactive= +2.53±0.61 (p<0.001) ; time =+0.042±0.0064 per day (p<0.001)

*It is a must that the authors improve the results section so the real impact of this research is realized.

*Why is there two Figure 2s? one in manuscript and one in Supplementary-

*Why is Pharmacokinetic capitalized “Predictive factors of Pharmacokinetic origin in

*I think that the following should not be in the manuscript text but be in the figures legend

“Probability to achieve CRP-based clinical remission status by week (2, 6 and 14) dur-281 ing induction by treatment group (standard and proactive) and baseline IFX Clearance 282 (median, with interquartile range). Estimates are: pop=2.49±0.78 (p=0.001); cl=-5.77±2.40 283 per L/day (p=0.016); proactive= +2.53±0.61 (p<0.001) ; time =+0.042±0.0064 per day (p<0.001)”. Also what the spacing and punctuation.

Panel A: PF of PK origin infusion 2; Panel B: PF of PK origin infusion 3; Panel C: PF of 288 PK origin infusion 4.

*Spacing in “*Model: logit(Probability of CRP based remission)= pop+covi*covi+… _” an spacing for titles are not the same in the tables and figures.

*figure 2 in manuscript is pixelated.

*The Supplementary file has two tables and two figures so the following is incorrect. “Supplementary Materials: The following supporting information can be downloaded at: 301 www.mdpi.com/xxx/s1, Figure S1 and Table S1

Discussion

Author’s guidelines indicate the following for discussion “Authors should discuss the results and how they can be interpreted in perspective of previous studies and of the working hypotheses. The findings and their implications should be discussed in the broadest context possible and limitations of the work highlighted. Future research directions may also be mentioned. This section may be combined with Results.”  The discussion is weak in the required aspects and should be improved without restating the results.

Generally speaking, the authors do not cite the results in the discussion and this makes it hard to provide credibility to statements made in the discussion. For example, authors stated “Yet, the data suggests that enhanced disease control”  and “As expected, higher IFX concentrations during induction predicted improved out-214 comes during maintenance, further adding to the body of evidence that optimal exposure 215 achieved early is important for patient outcomes.” but where is the data from the study….   Authors must cite back their evidence to support their statements.

Tense of the sentences should be checked. For example the following should be past tense….This was a retrospective exploratory analysis and our data suggest enhanced.    Please check whole manuscript.

*MDPI is committed to supporting open scientific exchange and enabling our authors to achieve best practices in sharing and archiving research data. We encourage all authors of articles published in MDPI journals to share their research data. Individual journal guidelines can be found at the journal ‘Instructions for Authors’ page. Data sharing policies concern the minimal dataset that supports the central findings of a published study. Generated data should be publicly available and cited in accordance with journal guidelines.

Why do the authors not want to share their data?

Difficult statements that need to be improved. There are too many to outline all of them, so authors must check all sentences again and correct the difficult language:

Our rationale was that lower Clearance, and thus improved retention of IFX-containing volume available for the neutralization of inflammatory burden, 196 would associate with improved disease control, simply the opposite of what is typically observed in the presence of active disease that eliminates IFX and worsens in the presence of immunization against IFX.

We noticed that baseline 204 Clearance associated with worse outcomes as already reported[9], an impact that was sig-205 nificant after adjusting for the treatment group where the potential negative impact of 206 higher baseline Clearance in the group of patients who received Proactive dosing was 207 minimized.

These data support the assertion that Clear-229 ance is as good as concentration, if not better, in associating with clinical outcomes, likely 230 due to an underlying degree of causality with improvements in clinical benefit; this sup-231 ports the clinical utility of PK measures for patients with IBD[3, 14, 33].

Lastly it is encouraged that the authors provide a pristine version of a manuscript in the next round.

Some areas need improvement. See comments above.

Author Response

  1. We have followed the recommendations of the reviewers and have removed the heads from the abstract.
  2. We have verified the punctuation, grammar and orthograph in the revised manuscript.
  3. We have modified the language as recommended by the reviewer.
  4. We have verified the brackets in the revised manuscript and modified the statement for clarity.
  5. We have verified the punctuation in the revised manuscript.
  6. The reference with CRP based clinical remission has been added
  7. We have followed the recommendation of the reviewers and have provided a concise and precise description of the results. The Figure has been moved to the main manuscript.
  8. We have carefully read through the result section and edited accordingly.
  9. We modified the legends of the table and figures. The Figure S2 has been moved as figure 2 in the revised manuscript.
  10. Figure 2 and S2 describes the performances of the PF of PK origin in combination in the main text, with the individual PF of PK origin (clearance and concentrations) in the supplementary materials
  11. We have removed the capital letters as appropriate.
  12. We respectfully believe that the description of the results in this section is appropriate to describe the results
  13. We have corrected the statement and respectfully believe that the corrections are appropriate.
  14. We have modified the discussion to reflect the limitation and future directions.
  15. The data are not shares due to data-sharing restrictions.

Round 2

Reviewer 2 Report

The revised version of the manuscript has indeed corrected numerous areas, but the second review of the manuscript has the following points to highlight, and reviewer encourages the authors to be diligent with their corrections:

Results

As per author’s guidelines: Results: Provide a concise and precise description of the experimental results, their interpretation as well as the experimental conclusions that can be drawn.

*Reviewer encourages the authors to be more explicit with the results. Have the tables follow the text when the initial mention is, and don’t have one sentence to describe a table or figure (i.e. figure 1). This does not do justice to the work in the manuscript.

There seems to be three tables in one in Table I. Would it seem more clear if they are all in one table?

Figures 1, 2, 3, and 4 are blurry (pixelated)

Figure 3 and 4 lack a legend. Why? Or is it in the text?  Legends should follow the title of the figure.

References are not in the correct structure. Please read the author’s guidelines and fix.

Where is the following graphical abstract? As in the following: We also acknowledge Andres Pere for graphical abstract.

Problems with brackets still. Square brackets are on the outer and curved brackets are in the inner, except for some rules like citations.

Has vastly improved.

Author Response

  1. We have followed the recommendation of the reviewer and have inserted the Table and Figures in the text when mentioned.
  2. We respectfully believe that Table I is the most appropriate representation of the data results.
  3. We have followed the recommendation of the reviewer and have recreated figures in high resolution.
  4. We have followed the recommendation of the reviewer and have added a legend to Figure 3 and 4
  5. The references have been formatted as per journal guidelines.
  6. The graphical abstract is provided in the front page.
  7. Updated graphical abstract attached following some feedback from Judy.
  8. We have checked parentheticals and brackets and did not find any errors in the attached version.